# Controlling the Degradation Rate of Biodegradable Mg–Zn-Mn Alloys for Orthopedic Applications by Electrophoretic Deposition of Hydroxyapatite Coating

**DOI:** 10.3390/ma13020263

**Published:** 2020-01-07

**Authors:** Iulian Antoniac, Florin Miculescu, Cosmin Cotrut, Anton Ficai, Julietta V. Rau, Elena Grosu, Aurora Antoniac, Camelia Tecu, Ioan Cristescu

**Affiliations:** 1Faculty of Materials Science and Engineering, Politehnica University of Bucharest, 313 Splaiul Independentei, District 6, 060042 Bucharest, Romania; antoniac.iulian@gmail.com (I.A.); f_miculescu@yahoo.com (F.M.); cosmin.cotrut@upb.ro (C.C.); elena_grosu@yahoo.com (E.G.); tecu_camelia@yahoo.com (C.T.); 2Faculty of Applied Chemistry and Materials Science, Politehnica University of Bucharest, 7 Gheorghe Polizu, District 1, 011061 Bucharest, Romania; anton.ficai@upb.ro; 3Istituto di Struttura della Materia, Consiglio Nazionale delle Ricerche (ISM-CNR), Via del Fosso del Cavaliere 100, 00133 Rome, Italy; giulietta.rau@artov.ism.cnr.it; 4Clinical Emergency Hospital Bucharest, Dept.Orthoped. & Traumatol, 8 Floreasca Ave, District 1, 014461 Bucharest, Romania; ioancristescu@yahoo.com

**Keywords:** magnesium alloys, Mg-Zn-Mn, hydroxyapatite, biodegradation, coating, electrophoretic deposition, simulated body fluid

## Abstract

Magnesium alloys as bioresorbable materials with good biocompatibility have raised a growing interest in the past years in temporary implant manufacturing, as they offer a steady resorption rate and optimal healing in the body. Magnesium exhibits tensile strength properties similar to those of natural bone, which determines its application in load-bearing mechanical medical devices. In this paper, we investigated the biodegradation rate of Mg-Zn-Mn biodegradable alloys (ZMX410 and ZM21) before and after coating them with hydroxyapatite (HAP) via the electrophoretic deposition method. The experimental samples were subjected to corrosion tests to observe the effect of HAP deposition on corrosion resistance and, implicitly, the rate of biodegradation of these in simulated environments. X-ray diffraction (XRD), scanning electron microscopy (SEM) and energy-dispersive spectroscopy (EDS) provided detailed information on the quality, structure, and morphology of the HAP coating. The obtained results demonstrate that coating of Mg-Zn-Mn alloys by HAP led to the improvement of corrosion resistance in simulated environments, and that the HAP coating could be used in order to control the biodegradation rate.

## 1. Introduction

Since the beginning of production, magnesium (Mg) has attracted attention in the industry owing to its low density, combined with mechanical properties similar to other metals, such as aluminium and steel, resulting in the improved production of light metal components with similar mechanical strength [1,2]. Due to its low mechanical properties, non-alloyed Mg is rarely used as a structural material, while the Mg-Zn-Mn (abbreviated ZM based on the main alloying elements) family of alloys represents a good combination of properties, such as high breaking strength, easy casting and good corrosion resistance [3]. Due to the lightweight and density value of Mg and Mg alloys between 1.74–2.0 g/cm^3^, similar to human bones (1.8–2.1 g/cm^3^) [4,5,6], they are used in implants for the short-term healing of patients. Magnesium, calcium, and zinc are minerals essential for human health, and therefore, these metallic components along with a small addition of rare elements are very appropriate for the delay of implant degradation [7,8]. 

The high degradation rate of Mg alloy implants in the human physiological environment results in the reduction of mechanical integrity of the implant before the bone tissues have sufficient time to heal. In order to reduce the degradation rate of Mg alloy implants, coating of the implant with bioceramics or biopolymers, using various coating methods represents a new area of research [8,9,10,11]. The coating of implants aims to improve their corrosion resistance and to maintain the mechanical strength. However, the degradation rate must not be overlooked since a too rapid degradation of implants can have a potential risk of inflammation due to excessive hydrogen production and loss of mechanical integrity of implant prior to healing [12,13]. Therefore, in order to be an effective biodegradable implant for orthopaedic applications, the degradation rate of Mg alloys should be sufficient for the healing process and new tissue growth to take place, in order to provide a natural support before the structural integrity of the implant weakens [14,15,16,17]. Calcium phosphate biomaterials are the most representative of the resorbable materials category, being the most preferred biocompatible synthetic substances used for bone implants. Compared to the majority of ceramic materials, phosphate-based implants have low compressive strength, yet comparable to the hard natural tissues. Exceptional biocompatibility of these materials can be attributed to the fact that they are capable of inducing a direct link with the bone since they have a similar chemical composition [18,19].

Hydroxyapatite Ca_10_(PO_4_)_6_(OH)_2_ (HAP), belonging to the calcium phosphate materials family, is a well-known bioceramic that has been extensively studied over the years for biomedical applications, being one of the major components of hard tissues: bones, teeth, and tendons. Due to its bioactivity, after implantation, HAP produces chemical species that support the implant’s adhesion to surrounding tissue by forming a functional connective structure [20,21]. 

Corrosion electrochemical reactions of Mg are presented in the Equations (1)–(3) [22,23,24,25,26,27].
Mg_(s)_→Mg^2+^_(aq)_ + 2e^−^        E = −2.37 V(1)
2H_2_O + 2e^−^→ H_2(g)_ + 2OH^−^_(aq)_   E = −0.83 V(2)

Biodegradable Mg alloys gradually corrode in vivo and completely dissolve in the body after carrying out the task of healing surrounding tissues. Therefore, the major component of biodegradable metal alloys would be the essential metal elements to be metabolized by the human body [28,29,30]. In order to increase the corrosion resistance of Mg alloys, the components made of Mg must be coated in a manner that inhibits the electrical contact between the metal surface and wet environment [31,32,33,34]. The deposition of HAP layers was accomplished by a large number of methods, including plasma spraying [35], spray coating [36], biomimetic methods [37], electrophoretic deposition [38], electrochemical deposition [39], electrospray deposition [40], and pulsed laser deposition [11]. In close relationship with the ability to select alloying elements and processing technologies, the surface modification techniques provide the opportunity to design a specific Mg alloy implant with mechanical properties and biodegradation profile that can be tailored for the specific orthopaedic applications [41,42,43,44]. Until now, only some Mg alloys, such as AM50, AZ31, AZ 61 [45], AZ 91 [46], MCZ, MCZS, ZE41A [47,48,49], LAE 442, and WE43 [50], were developed to be used as implants. Scarce information is available about other Mg alloys, such as, for example, ZM 21, whose degradability in vitro was investigated in [51], and ZMX 410. This encouraged us to study these materials for their future use in biodegradable implants. 

In this article, we present the biodegradation behaviour of two Mg-Zn-Mn based alloys (ZM21 and ZMX410), uncoated and coated with HAP. Simulated body fluid (SBF) was used as a simulated environment. The biodegradation behaviour was highlighted by mass loss evaluation, hydrogen release and corrosion rate determination. The surface morphology and biodegradation products that occurred on the surface of Mg-Zn-Mn alloys upon experiments, were evaluated by Fourier-transform infrared spectroscopy (FTIR), by scanning electron microscope (SEM) coupled with the energy dispersion spectrometer (EDS) to reveal elemental compositions, and by X-ray diffraction (XRD) to confirm the nature of reaction products. 

In order to minimize corrosion, and to maintain mechanical integrity and interfacial biocompatibility, we performed surface modifications by deposing the HAP layer by the electrophoresis method [52]. Electrophoretic deposition (EPD) is one of the colloidal processes in ceramic production. It has the advantage of forming layers in a short time, requiring a simple apparatus. The deposition can be done on planar, cylindrical, or any other shape surfaces with a minor change of the electrode’s design and positioning. In particular, despite the fact that the EPD is a wet process, it can control the thickness and morphology of the deposited film by simply changing the deposition time and the applied potential. Powder particles loaded, dispersed, or suspended in a liquid medium are attracted and deposited on a conductive substrate with an opposite load on the application of an electric field of direct current. Bioactive ceramic coatings play a double role because they prevent the release of noxious metal ions from the metallic substrate (increasing the corrosion resistance of implants) and restore the bioactivity of the implant surface. HAP coating can provide a stable interface between orthopaedic implant and bone. Extensive studies have demonstrated that EPD is a particularly attractive technique for depositing HAP on a metal substrate, also allowing its impregnation in porous metallic structures, e.g., scaffolds for bone regeneration. With respect to the thickness of the deposited layer achieved by EPD, it can vary from 0.1 μm to more than 100 μm [53,54].

## 2. Materials and Methods 

### 2.1. Magnesium Alloys and Hydroxyapatite

The main materials, Mg alloys ZM21, ZMX410, and HAP, used in this study, result from raw materials that required intermediate processing steps to be used further in the EPD process. ZM21 and ZMX410 alloys were purchased from the DSM Company (Beersheva, Israel). Their chemical composition, presented in Table 1, consists mainly in non-ferrous metals: Mg as major component together with the presence of zinc (Zn), yttrium (Y), manganese (Mn), and calcium (Ca) in very low mass concentration, according to technical data sheets. 

The raw materials used to prepare the HAP solution for the coating process were HAP and isopropyl alcohol. HAP was obtained from bovine bone, and the solvent isopropyl alcohol concentration 99.70% was purchased from the S.C. Chemical Company S.A. Simulated body fluid (SBF), prepared in accordance with the Kokubo’s recipe [55], was used as simulated environment in analyses of the corrosion behaviour of the ZM21, ZMX410 alloys uncoated and coated with HAP.

In order to perform the corrosion resistance experiments, the ZM21, ZMX410 alloy samples were prepared by going through several stages, namely:-obtaining Mg alloy specimens and evaluating their surface properties;-preparing HAP solution;-deposing HAP on the ZM21, ZMX410 alloys surface by EPD.

Specimens of ZM21, ZMX410 alloys were prepared by cutting through wire erosion, into square shape with dimensions of 12 × 12 × 2 mm, by means of the Fanuc Series 18i apparatus, which can cut any conductive material using an electric conductive wire with very small diameter. The device consists of a control panel and work area. After cutting specimens, their grinding was performed on metallographic paper with granulation of 1000 and 1200, followed by the polishing on metallographic paper with granulation of 2500 together with a diamond particle (average 1 μm) paste, with the StruersTegramin 25 apparatus.

#### 2.1.1. Preparation of Hydroxyapatite Solution 

Bovine HAP was processed through the Ball Milling wet method using the BM-0902 apparatus. The solution to be used in EPD, composed of 80 wt% isopropyl alcohol and of 10 wt% bovine HAP powder as raw materials, was prepared by mechanical stirring with 400 rpm at a temperature of 25 °C for 30 min using the AREX - Heating Magnetic Stirrer, followed by ultrasound process for further 30 min to achieve a more uniform powder distribution without significant changes in the primary HAP structure. Ultrasonic treatment was performed with a Bandelin Sonorex apparatus of 20 kHz and 130 W. After ultrasonic treatment during period of 30 min at ultrasound intensity of 0.65 W/cm^3^ corresponding to power density, the HAP solution was stored at 20 °C before being analyzed.

#### 2.1.2. Deposition of Hydroxyapatite on the Magnesium Alloys Surface by Electrophorethic Method 

The coating of the ZMX410 and ZM21 alloys with HAP was performed using the Laboratory DC power supply, Model GPR-3510HD Corp. apparatus powered by a 37 V voltage. Each of the alloys were immersed into the HAP solution one at a time, the deposition process lasted 5 min.

### 2.2. Characterization Methods

#### 2.2.1. Scanning Electron Microscopy coupled with Energy Dispersive Spectroscopy

SEM-EDS was used to characterize the surface morphology and to determine concentration of the constituent elements of coated and uncoated samples before and after corrosion assessments in SBF.In order to evaluate the coating uniformity over the magnesium alloys substrates, SEM was used to measured, at various locations, coating thickness. The experimental samples were prepared using argon ion beam in order to see the cross-section of the coating layer.

#### 2.2.2. X-ray Diffraction

XRD measurements were carried out using an X-Pert PRO Materials Research Diffractometers (MRD) Panalytical X-Pert PRO diffractometer (UK), working in the Bragg-Brentano symmetric geometry. The XRD measurements were performed at diffraction angle 2θ within range of 8–80°.

#### 2.2.3. Fourier-Transform Infrared Spectroscopy

FTIR was performed before and after corrosion assessment by a Nicolet iS50 FTIR spectrometer (USA), equipped with a deuterated triglycinesulfate (DTGS) detector, which provides information with a high sensitivity in the range of 4000 cm^−1^ to 400 cm^−1^. The spectra were obtained in the attenuated total reflection (ATR) mode (diamond crystal) over the range of 400–4000 cm^−1^, with a resolution of 4 cm^−1^ and 32 spectra were co-added to obtain the high quality spectra with limited necessity of smoothing. IR microscopy was performed by using a Thermo FTIR Nicolet iN10 MX microscope (USA); the spectra were recorded in the reflection mode in the range of 675–4000 cm^−1^, with a resolution of 4 cm^−1^ and collection time of 3 s.

#### 2.2.4. Corrosion Resistance of the Magnesium Alloys in the Simulated Media

The corrosion resistance assessment of ZM21 and ZMX410 alloys was performed in SBF, Kokubo solution. The SBF was prepared in the laboratory, according to the protocol proposed by Kokubo [51]. The experiments of biodegradation were conducted by a simple immersion test and by electrochemical method.

##### Immersion Test

Immersion test consisted in immersing of uncoated samples in SBF and observing and quantifying the changes occurring in the mass of the ZM21 and ZMX410 alloy materials. Corrosion behaviour was analysed through corrosion rate assessment via the determination of hydrogen emission, evaluation of pH, and mass loss of the samples during experiments. Also, the corrosion behaviour of the ZM21 and ZMX410 samples was analysed by a daily registration of mass loss assessment.

• Hydrogen Release Rate

The test for determining the hydrogen release rate in simulated media for the Mg alloys was developed by Song et al. [56]. This test is based on the fact that during the corrosion process hydrogen is produced according to the following reaction (Equation (3)).
Mg_(s)_ + 2H_2_O_(aq)_→ Mg(OH)_2(s)_ + H_2(g)_(3)

The process is relatively simple and consists in measuring the volume of hydrogen released during immersion of the Mg alloy in simulated environment, this being proportional to the amount of dissolved Mg alloy. The setup for determination of the volume of degassed hydrogen was composed of a crystallizer, a funnel and a graduated cylinder. Experimentations were carried out at room temperature.

• Behaviour of the Simulated Media

The measurements of the pH values of the SBF were performed daily, using the apparatus HI 2210 pH Meter from HANNA Instruments, Germany.

• Mass Loss and the Rate Corrosion of the Mg Alloy Samples

The experiment consisted in immersion of the ZM21 and ZMX410 samples in the polyethylene dishes with 50 cm^3^ of SBF and sinking the dishes into a thermostatic bath type Thermo-circulator from LabTech (Italy), in which water is maintained at a temperature of 37 °C similar to the human body. The corrosion behaviour of the samples was observed by examining the surface modifications and by weighing them after removing from SBF and subsequent drying. The ZM21 and ZMX410 samples were weighed with the balance type WLC 2/A2 produced by RADWAG Balances and Scales, at the beginning and daily at the same hour, for 14 days.

The values recorded for the mass of samples were used to calculate the mass loss (Δm (%)) during the experiment, using the following formula [57]:(4)$$\Delta m[\%]=\frac{M_{f}-M_{i}}{M_{i}}\times 100$$
where Δm = the mass loss measured between every two recorded mass of the samples; M_i_ = the initial mass value recorded at the beginning of the experiment; M_f_ = the final mass value at the end of the experiment.

The corrosion behaviour was assessed by calculating the rate of corrosion, according to the following formula [58]:(5)$$CR=\frac{8.76\times 10^{4}\times \Delta g}{A\times t\times \rho }$$
where *CR* = corrosion rate; Δ*g* = modification of sample mass (g); *A* = the area of the initial surface exposed to corrosion; *t* = immersion time (h); *ρ* = alloy density.

##### Evaluation of Electrochemical Corrosion Behaviour of the Uncoated and Hydroxyapatite Coated Specimens

The corrosion resistance of the samples was examined on the basis of several evaluation criteria. The corrosion resistance evaluation tests were performed using a Potentiostat/Galvanostat (PARSTAT 4000 model, Princeton Applied Research, Oak Ridge, TN, USA), and the potentiodynamic curves were obtained using Versa Studio software. Electrochemical tests using the linear polarization technique were performed according to the ASTM G5-94 (2011). To perform the tests, a corrosion cell, consisting of a calomel saturated electrode (SCE) as a reference electrode, a platinum electrode as a recording electrode, and a working electrode consisted of the investigated samples (uncoated and HAP coated Mg alloys) were used. Corrosion tests were performed in SBF at the temperature of 37 ± 0.5 °C using a heating and recirculation bath model CW-05G produced by the Jeio Tech. The corrosion resistance was determined by the Tafel technique, which consists in drawing the linear polarization curves involving the following steps:open circuit potential (E_oc_) measurement over 1 h;plotting the linear polarization curves from –250 mV (vs. OCP) to + 250 mV (vs. OCP) - Tafel curves, with a scan rate of 1 mV/s.

From the Tafel curves [59], the following parameters were determined:open circuit potential (E_oc_);corrosion potential (E_corr_);the density of the corrosion current (i_corr_);the slope of the cathodic curve (β_c_);the slope of the anodic curve (β_a_).

Using the parameters determined from the Tafel curves, the following characteristic parameters for the corrosion resistance of the investigated samples were calculated:corrosion rate (CR);polarization resistance (R_p_);corrosive attack efficiency (P_e_).

Calculation of the corrosion rate was performed according to ASTM G102-89 (2015) using the following formula:(6)$$\mathrm{CR}=K_{i}\frac{i_{\mathrm{corr}}}{\rho }\mathrm{EW}$$
where CR refers to the corrosion rate; K_i_= 3.27 810^−3^; ρ is the density of the material; i_corr_ denotes the density of corrosion current; EW refers to the equivalent weight.

The polarization resistance was performed according to ASTM G59-97 (2014) using the formula:
(7)$$R_{p}=\frac{1}{2.3}\cdot \frac{\beta_{a}\beta_{c}}{\beta_{a}+\beta_{c}}\cdot \frac{1}{i_{\mathrm{corr}}}$$
where:

*β_a_* = the slope of the anode curve;

*β_c_* = the slope of the cathode curve;

i_corr_ = density of corrosion current.

The protection against corrosive attack was calculated using the formula:
(8)$$P_{e}=(1-\frac{i_{corr,layer}}{i_{corr,substrate}})\cdot 100$$
where *i_corr,layer_* refers to the density of corrosion of the layer and *i_corr,substrate_* is the density of the corrosion current of the substrate.

#### 2.2.5. Contact Angle Determination

The measurements of the surface contact angles were performed on the KRÜSS DSA30 Drop Shape Analysis System using the Fowkes method [60]. This method is based on a combination of the knowledge of Fowkes on the one hand and Owens, Wendt, Rabel, and Kaelble on the other, as Fowkes initially determined only the disperse part and the latter determined both the components of the surface energy. In the first step the disperse part of the surface energy of the solid is calculated by making contact angle measurements with at least one purely dispersed liquid. By combination of the surface tension equation of Fowkes for the disperse part of the interactions:(9)$$\gamma_{sl}=\sigma_{s}+\sigma_{l}-2\sqrt{\sigma_{s}^{D}\cdot \sigma_{l}^{D}}$$
with the Young’s equation:(10)$$\sigma_{s}=\gamma_{sl}+\sigma_{l}\cdot \cos\theta$$
the following equation for the contact angle is obtained after transposition:(11)$$\cos\theta =2\sqrt{\sigma_{s}^{D}}\cdot \frac{1}{\sqrt{\sigma_{l}^{D}}}\;-\;1$$

Further, based upon the general equation for a straight line,
y = mx + b,(12)
cosθ is then plotted against the term 1σlD. The term $2\sqrt{\sigma_{s}^{D}}$ can be determined from the slope m.

## 3. Results and Discussion

### 3.1. Scanning Electron Microscopy (SEM) Coupled with Energy-Dispersive Spectroscopy (EDS) Results

SEM micrographs were analysed to assess the surface morphology of the ZM21 and ZMX410 alloys and the homogeneity and uniformity of the HAP coatings. In Figure 1, the micrographs corresponding to the plain view of the uncoated ZM21 and ZMX410 samples are presented. 

SEM micrographs of plain views present uniform morphology at the surface both in the ZM21 and ZMX410 alloys. The cross section micrographs of the HAP coated samples, presented in Figure 2, at different magnifications, offer evaluation of the surface morphology of the HAP coatings. In Figure 2a, a plain view of HAP coating on the ZM21 alloy is shown, characterized by a uniform and porous morphology, and in (b) a HAP layer of approximately 15–16 μm thickness was observed in the micrograph at 3000×.

In Figure 2c, the surface of coated ZMX410 specimen exhibits homogeneity and a uniform aspect over the entire surface of the sample. The morphology of this sample presents a porous structure. Thickness of the hydroxyapatite layer, as presented in Figure 2d magnified at 3000×, was 15 µm.

Elemental composition of the ZM21 and ZMX410 samples coated with HAP was determined by the EDS method. Figure 3a presents the morphology of the coated ZM21 surface and Figure 3b exhibits its elemental composition, characterized by the presence of the main elements of the alloy and HAP.

Figure 4 presents the EDS spectra registered in some points highlighted in the SEM micrograph of the coated ZMX410 sample (Figure 4a). EDS spectra contained data that refer to globular particles presented in Figure 4b (blue arrow in the Figure 4a), surface (Figure 4c; white arrow in the Figure 4a) and compounds at the grain boundary (Figure 4d; red arrow in the Figure 4a).

According to the Energy-Dispersive Spectroscopy (EDS) measurement, we could conclude that Ca and Zn are distributed mainly at the grain boundary and in some globular particles. 

### 3.2. XRD Results

Prior to performing the corrosion experiments on the uncoated and HAP coated ZM21 and ZMX410 samples, they were subjected to the XRD analysis. According to the XRD diffractograms, the coating layers are composed of pure HAP. Indeed, a diffraction peak at about 2θ = 33°, corresponding to the semi-crystalline HAP (according to the ICDD9-432), was registered. The other constituent phases of the ZM21 and ZMX410 alloys were not present in the XRD patterns because their amounts in the bulk material were below the detection limit of the instrument. In particular, the XRD diffractograms of the uncoated and HA coated ZM21 alloys, presented in Figure 5a,b, exhibit two main peaks at 2θ = 32.37° and 2θ = 47.86°, corresponding to Mg, and at 2θ = 47.79°, corresponding to Mn. Diffractogram (b) also highlights the main at 2θ = 33° along with broad peaks at 2θ = 25.95°, 2θ = 31.82°, 2θ = 32.37°, 2θ = 34.11°, indicating the presence of HAP on the ZM21 surface. Other peaks at 2θ = 32.36° and 2θ = 34.48° were attributed to Mg, and at 2θ = 31.27° and 2θ = 33° to Mn.

The diffractogram of the ZMX410 alloy (Figure 6a) presents broad peaks corresponding to Mg, with the most important at 2θ = 68.74° and 2θ = 70.2° (high intensity). Other peaks belonging to some compounds on the uncoated metal surface are those at 2θ = 42.93°, corresponding to manganese zinc (MnZn_3_)_0.5_, 2θ = 34.65° and 2θ = 70.2°, corresponding to calcium magnesium CaMg_2_ and 2θ = 34.45° and 2θ = 57.54°, corresponding to magnesium zinc Mg_51_Zn_20_. In the diffractogram (b), several peaks with the most intense at 2θ = 33° account for the presence of HAP on the ZMX410 alloy surface. Peaks with very low intensity, corresponding to Mg, are presented in the diffractogram, the most important being at 2θ = 32.25° and 2θ = 34.45°.

### 3.3. FTIR Results

According the results of the FTIR spectroscopy, the deposited coatings are composed of HAP (Figure 7). Indeed, the main peaks assignable to HAP are: ~567 and ~600 cm^−1^ corresponding to the ν_4_(PO_4_) mode, ~630 cm^−1^ assigned to the HO liberation, 1023/1027 and 1087 cm^−1^ assigned to the ν_3_(PO_4_) mode [61]. According to both the spectra presented in Figure 7, it can be concluded that a slight carbonatation occurs during the deposition process, confirmed by the presence of the characteristic peaks at ~870, ~1410, and ~1455 cm^−1^, while the content of the brushite (CaHPO_4_ × 2H_2_O) phase is comparable with the HAP, the band at ~960 cm^−1^ has a low intensity. The presence of carbonated apatite is beneficial because it assures a better biocompatibility and a higher resorption rate comparing to the pure HAP [62]. 

Moreover, the FTIR chemigram maps were recorded for the coated samples, in order to evaluate the coating’s morphology and homogeneity (Figure 8 and Figure 9). For a better visualization of the coating, some defects were created, and the FTIR maps of an area about 1100 × 800 µm^2^ were recorded close to these defects (for the coated ZM21 sample a bright strip can be visualized on the right side of the dark field image; Figure 8). Even if hill–valley morphology can be identified onthe surface of the coated ZM 21 alloy, the coating is continuous, and therefore the electrophoretic deposition confers the premises to be used as deposition method. The evaluation was carried out at the maximum of the main apatite peaks. The most intense band of HAP, centred at ~1020 cm^−1^ was selected for the evaluation of the HAP deposition. One can see that the map registered with a reference band ~1020 cm^−1^ is very similar to that with 965 cm^−1^, which confirms that brushite is homogeneously dispersed within the HAP coating. Carbonate presence, according to the maps recorded with ~870, 1422 or 1458 cm^−1^, is highlighted by a quite homogeneous distribution with limited differences comparing with the map of the HAP, as well as a very good similarity with the map recorded with the reference bands of the OH group (centred at 3645 or 3565 cm^−1^). Mostly, the red, green, and blue colour shifts are assigned to the hill–valley topography. 

The sample ZMX 410 highlights a homogeneous coating of HAP as major component as well as traces of brushite and carbonated apatite. The deposition is quite homogeneous from both a morphological and compositional point of view (especially the distribution of HAP, HAP-CO_3_ and CaHPO_4_ × 2H_2_O), as can be observed in the three maps recorded with reference characteristics wavelengths of the calcium phosphate species (Figure 9).

### 3.4. Corrosion Behaviour of the Mg Alloys in the SBF

#### 3.4.1. Immersion Test

During the immersion test, a fast and accentuated degradation of ZMX410 alloy, as a consequence of the chemical reaction between SBF components and calcium from metallic matrix was observed. This led to the formation of solid microphases together with the release of important quantities of hydrogen, and small pieces detached from the bulk of the samples. The hydrogen release from the ZMX410 sample occurred at high rate during the entire experiment, concomitant with the fast fragmentation and total degradation of the sample. Instead, the ZM21 material did not degrade during the immersion, so that the amount of hydrogen released was very small, and the rate almost constant during the entire experiment (Figure 10a). The pH values of both the ZM21 and ZMX410 alloys in SBF, expressed through value domains of 7–9.8, rapidly increased during all immersion times (see Figure 10b).

Corrosion rates for the ZM21 and ZMX410 samples, calculated according to the Equation (5), are presented in Table 2.

Degradation of the ZM21 and ZMX410 samples was investigated also through the determination of mass loss during immersion in the SBF (Figure 11). The composition of metallic alloy matrix, the size of grains, thermal treatment, and morphology has an important influence on the degradation by corrosion of the Mg alloys [63]. Upon the immersion in SBF, weighing of samples revealed interesting behaviour during the first 24 h. The ZMX410 adsorbed a small quantity of the SBF due to the adsorption of Cl^−^ ions. Then micro-cracks within the material appeared, and exfoliations occurred until the total degradation after 72 h. This phenomenon can be explained by the extension of the Mg^2+^ ions solubility and the formation of Mg(OH)_2_, together with hydrogen evolution and rising pH values of SBF [64]. The ZM21 alloy did not interact with SBF, so that the mass remained almost constant.

The appearance and dimensions of the samples during the SBF immersion underwent changes daily, accompanied by the development of microcracks, as can be seen in Figure 12.

#### 3.4.2. Evaluation of Electrochemical Corrosion Behavior of the Uncoated and HAP Coated Specimens

##### Corrosion Resistance

The variations of open circuit potential (E_oc_) corresponding to uncoated and HAP coated alloys in the SBF are presented in Figure 13, and the Tafel curves are shown in Figure 14. In order to evaluate the corrosion behavior, we used two coated samples for each type of the Mg alloys (coated ZM21: ZM21-1 and ZM21-2; coated ZM410: ZM410-1 and ZM410-2).

In Table 3, the main parameters determined during the electrochemical corrosion process are shown.

The corrosion potential (E_corr_) with the more electropositive values reveals a better corrosion behaviour. Electrochemical measurements showed that the corrosion potential values obtained for the uncoated and HAP coated ZM21 alloys are approximately equal, with only quite a small difference: 18 mV for the ZM 21-1 and 23 mV for the ZM 21-2. Even though the corrosion potential of the HAP coated ZM21 alloys are less electropositive (with just a few mV) compared to the Mg alloy substrate, an overview of all electrochemical parameters must be made prior to making a conclusion. It is known that a low corrosion current density (i_corr_) indicates a good corrosion resistance. Therefore, if we take this criterion into account, in the case of the SBF experiments, we see a decrease in the corrosion density of the HAP coated ZM 21 alloy demonstrating a better corrosion resistance than the uncoated one. The current density values for the HAP coated ZMX 410 alloy did not result in a decrease of the value of this electrochemical parameter, being the difference of a few μA/cm^2^. It is known that a high polarization resistance accounts for a good corrosion behaviour of a material, and a low value for a worse corrosion behaviour. Thus, it is observed that the HAP-coated ZM 21 alloys exhibit higher values than the uncoated ZM 21 alloy. Further, in the case of the ZMX 410 alloy, after coating with HAP, the polarization resistance exhibits improvements. After calculating the corrosion rate (CR) of the Mg alloys subjected to the corrosion tests in SBF, one can see that the values of both alloys are approximately equal, the difference being 0.51 mm/year, in favour of the ZMX 410 alloy. From the point of view of the efficiency against corrosive attack (P_e_), it can be concluded that in the case of ZM 21 alloy, the coating with HAP leads to a protection of approximately 40.6% (the average of the two investigated samples) and in case of the ZMX 410 alloy, a protection against corrosive attack of 4.23% is registered.

##### Surface Analysis by SEM- EDS of the Samples after Corrosion Experiments

SEM analysis coupled with EDS of the ZM21 and ZMX410 alloys indicated the formation of new phases or microconstituents after the corrosion experiments. As observed in Figure 15, the SEM micrograph of the coated ZM21 samples showed the presence of the HAP layer on the surface that protected the alloy against the electrochemical corrosion in SBF, demonstrating the stability of the coating. EDS analysis of the coated ZM21 alloy after electrochemical corrosion in SBF showed that the material contains all the alloy constituent elements of the HAP and SBF, namely Mg, Mn, Zn, and additionally, O, Na, P, Cl, and Ca. EDS spectra demonstrate the presence of few pitting corroded zones (Figure 15b) of the metallic surface, and majority of uncorroded zones (Figure 15c), where Ca/P of 1.65 ratio reveals the presence of an important quantity of HAP.

In the case of the coated ZMX410 alloy, electrochemically corroded in the SBF solution, the SEM micrograph presented in Figure 16a, exhibits crevice corroded zones. The EDS spectrum (Figure 16b) highlights the important amounts of the HAP elemental components, and also some traces of the SBF elemental components.

##### XRD Analysis of the Samples after Corrosion Experiments

The evolution of phases during the corrosion process investigated by the XRD assessment is characterized mainly by the peaks belonging to the Mg phase, HAP, Mn, and the compounds resulting from chemical reactions between the elements of the alloys and constituents of the simulated media, as can be observed in the diffractograms in Figure 17 and Figure 18. After the electrochemical and immersion corrosion of the ZM21 alloy experiments in the SBF, the XRD patterns in Figure 17a exhibit the main peaks corresponding to Mg at 2θ = 36.85° and 2θ = 68.66° with highest intensity. Instead, the diffractogram from Figure 17b depicts layer integrity of the HAP according to the main peak at 2θ = 68.69°, together with some microconstituents - chemical reaction products, such as: manganese yttrium oxide (YMnO_2.8_) with the main peaks at 2θ = 34.63° and 2θ = 36.73°, Mn with corresponding peaks at 2θ = 34.63° and 2θ = 68.69°, and calcium chloride phosphate (Ca_5_(PO_4_)_3_Cl) with the corresponding main peaks at 2θ = 34.63° and 2θ = 48.07°. Diffractogram from Figure 17c exhibits the presence of the main peak at 2θ = 47.99° of high intensity and 2θ = 32.21° of lower intensity, corresponding to Mg, and peaks at 2θ = 47.99° (high intensity) and 2θ = 32.21°, corresponding to Mn. Further, diffractogram from Figure 17c reveals that corrosion of the ZM21 occurred during the immersion test and some amounts of the alloy have chemically reacted with the SBF, generating magnesium phosphate hydrate (Mg_3_(PO_4_)_2_∙10H_2_O) represented by the peaks at 2θ = 12.25° and 2θ = 36.73°.

XRD patterns of the ZMX410 alloy electrochemically corroded in the SBF (Figure 18) exhibit for the uncoated sample in Figure 18a three main peaks at 2θ = 32.24°, 2θ=36.69° and 2θ=70.15°, corresponding to Mg, some low intensity peaks at 2θ = 42.2° and 2θ = 70.87°, corresponding to Mn, and peaks at 2θ = 63.18° and 2θ = 67.42°, corresponding to magnesium hydroxide. Diffractogram from Figure 18b exhibits peaks at 2θ = 32.29° and 2θ = 47.99°, corresponding to Mg, and its oxides at 2θ = 36.79°, hydroxide compound peaks at 2θ = 57.62° and 2θ = 63.18°, and Ca peak at 2θ = 35.67°. Diffractogram from Figure 18c exhibits only the peaks at 2θ = 18.75° and 2θ = 38.17°, corresponding to magnesium hydroxide compounds present in the powder as a consequence of corrosion by the immersion of the ZMX410 alloy in the SBF solution.

##### FTIR Analysis of the Samples after Corrosion Experiments

The coated ZM 21 sample was also exposed to the SBF, in order to evaluate its in vitro behavior. According to the obtained spectra, it can be concluded that these coated samples are bioactive and carbonated during the immersion. The surface change is heterogeneous, as it was demonstrated by the FTIR microscopy in both the video and IR imaging mode (Figure 19). In the video mode, light and dark areas indicate heterogeneities, while in the imaging mode, according to the maps recorded with the 1026 cm^−1^ reference band (specific to HAP) and with 1420 cm^−1^ (specific to carbonate) important heterogeneities can be observed, these areas being of up to 100 µm^2^. It is also important to mention that carbonatation leads to an increase in the hydrophilicity of the surface, as the maps recorded at a characteristic wavelength of free water (3366 cm^−1^) were very similar to that of carbonate (1420 cm^−1^).

The coated ZMX 410 sample was exposed to the SBF, and behaviour similar to that registered for the coated ZM21 was observed. According to the FTIR spectra, it can conclude that carbonatation and hydration occur. The carbonatation process is heterogeneous; in this case the size of the carbonated areas can reach even 300–500 µm, their shape being irregular. Also, it is worth mentioning that the compositional homogeneity of the coated ZMX410 is maintained after the SBF immersion (Figure 20).

### 3.5. Contact Angle 

The application of the ZM21 and ZMX410 alloys in the medical field requires the improvement of surface properties to ensure their biocompatibility and good interactions at the biological environment–implant interface. Increased wettability properties contribute to optimizing the response of implant material to physiological condition [65]. The hydrophilic/hydrophobic character is influenced by material composition and treatment, and also by homogeneity of the surface [66]. Material composition and surface roughness have an important influence on the contact angle and wettability of a surface. The smaller values of contact angles (CA) obtained during in the analysis and water behavior in contact with the uncoated and coated ZM21 and ZMX410 alloys reflect the surface properties of hydrophilic character, good adherence, and thin film formation ability.

The contact angle values of both the uncoated ZM21 and ZMX410 alloys determined in contact with water are maintained at relatively low values of 72.9–89.6 degrees, exhibiting the hydrophilic character and implicit good wettability properties (Figure 21). The water adhesion on the surfaces is attributed to different Van der Waals attraction forces [67]. This characteristic is determined by the alloy composition, surface roughness and fabricating process and exhibits limited-term integrity in simulated media aqueous solutions. After coating the Mg alloys with HAP, the values of contact changed, decreasing by 38% for the coated ZM21 alloy and by 81.1% in the case of the coated ZMX410, resulting in the improvement of hydrophilic wettability properties. The evolution of water drops on the coated and uncoated Mg surfaces is demonstrated in Figure 22.

## 4. Conclusions

Electrophoretic deposition resulted in a coating with a uniform and continuous HAP layer of approximate thickness of 15–16 μm. XRD analyses revealed that pure HAP was deposited, and no other phases or structural changes occurred. The elemental composition of Mg-Zn-Mn alloys determined by the EDS method was in accordance with the results obtained by the XRD and FTIR analyses. Therefore, microphases generated during the corrosion experiments consisted mainly of oxides and hydroxides of Mg or Mn. Also, the presence of HAP on the surface of experimental samples made by ZM21 and ZMX410 alloys after electrochemical corrosion experiments was confirmed. After electrochemical corrosion and immersion test, microphases formed on the surfaces were revealed by the XRD analyses, offering good understanding of degradation mechanisms. Experimental determinations of the corrosion resistance during the electrochemical and immersion tests have shown that our investigated Mg alloys exhibit a different response. The ZM21 alloy has a low corrosion rate in the SBF, of 1.01 mm/year at immersion experiments and 6.85 mm/year at electrochemical corrosion. The ZMX410 alloy demonstrated fast degradation with corrosion rate of 39.77 mm/year when immersed in the SBF and 6.34 mm/year after electrochemical experiments in the SBF. The two HAP coated ZM21 and ZMX410 alloys presented good corrosion response during the electrochemical experiments. Based on the experimental results obtained, we consider that the degradation rate of biodegradable Mg–Zn-Mn alloys for temporary orthopedic applications could be controlled by the electrophoretic deposition of hydroxyapatite coating [11,18,68,69]. 

## Figures and Tables

**Figure 1 materials-13-00263-f001:**
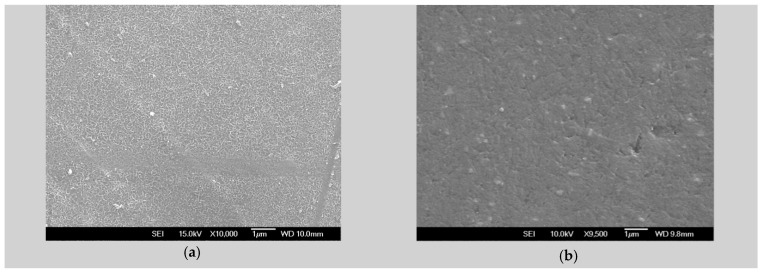
SEM micrographs of uncoated alloysamples: (**a**) ZM21 plain view, magnification 10,000×; (**b**) ZMX410 plain view, magnification 9500×.

**Figure 2 materials-13-00263-f002:**
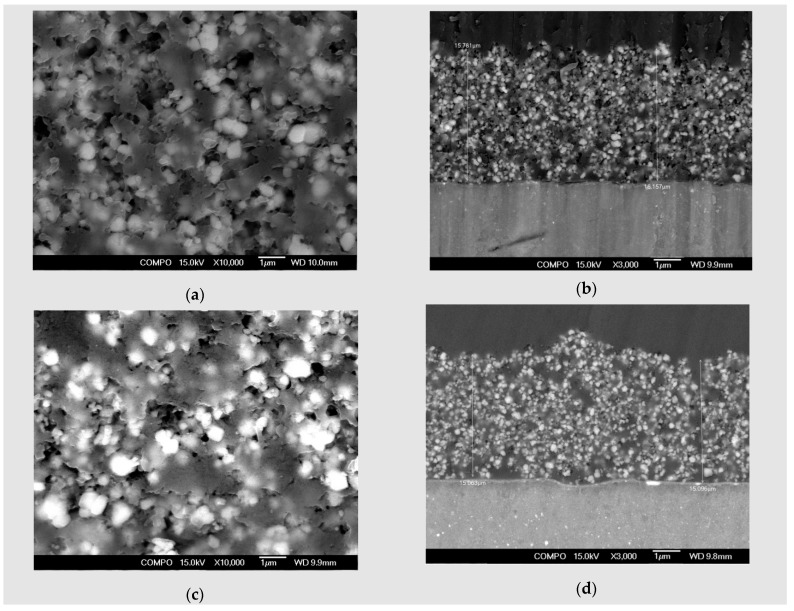
SEM micrographs of Mg alloys coated with HAP: (**a**) coated ZM21 plain view, magnification 10,000×; (**b**) coated ZM21 cross section view, magnification 3000×; (**c**) coated ZMX410 plain view, magnification 10,000×; (**d**) coated ZMX410 cross section view, magnification 3000×.

**Figure 3 materials-13-00263-f003:**
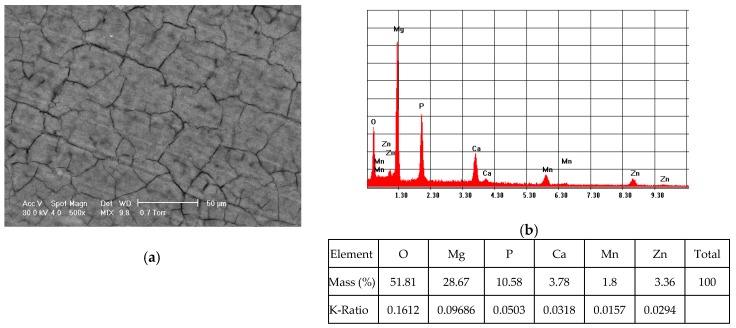
Representative SEM images (magnification 500×) of the coated ZM21 sample (**a**) and corresponding EDS spectrum (**b**).

**Figure 4 materials-13-00263-f004:**
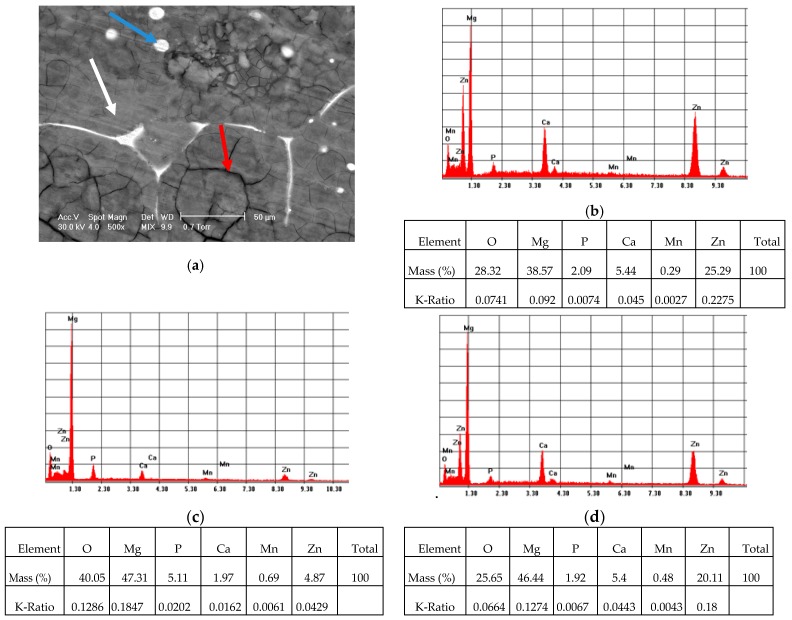
Representative SEM images (magnification 500×) of the coated ZMX410 sample (**a**) and corresponding Energy-Dispersive Spectroscopy (EDS) spectrum: (**b**) globular particles (blue arrow); (**c**) surface (white arrow); (**d**) compounds at the grain boundary (red arrow).

**Figure 5 materials-13-00263-f005:**
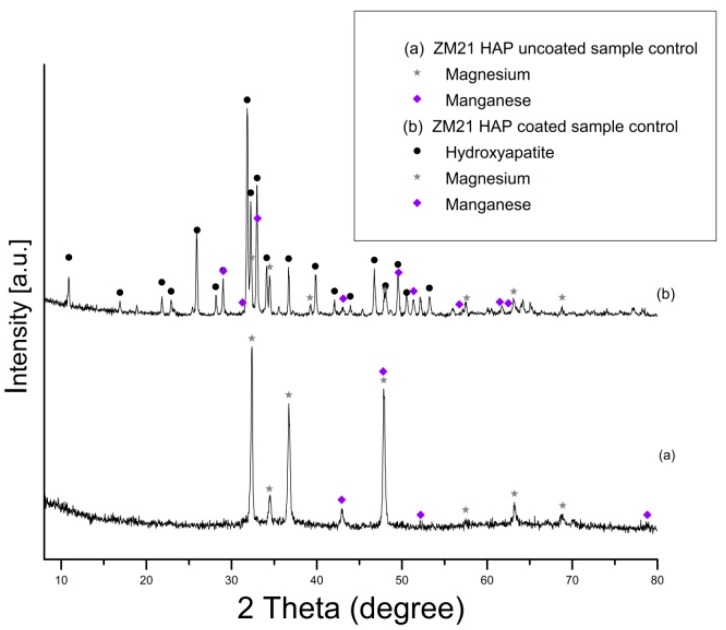
XRD patterns of uncoated (**a**) and hydroxyapatite (HAP) coated (**b**) ZM21 alloy (bullets indicate the peaks corresponding to diffraction angles of the main phases).

**Figure 6 materials-13-00263-f006:**
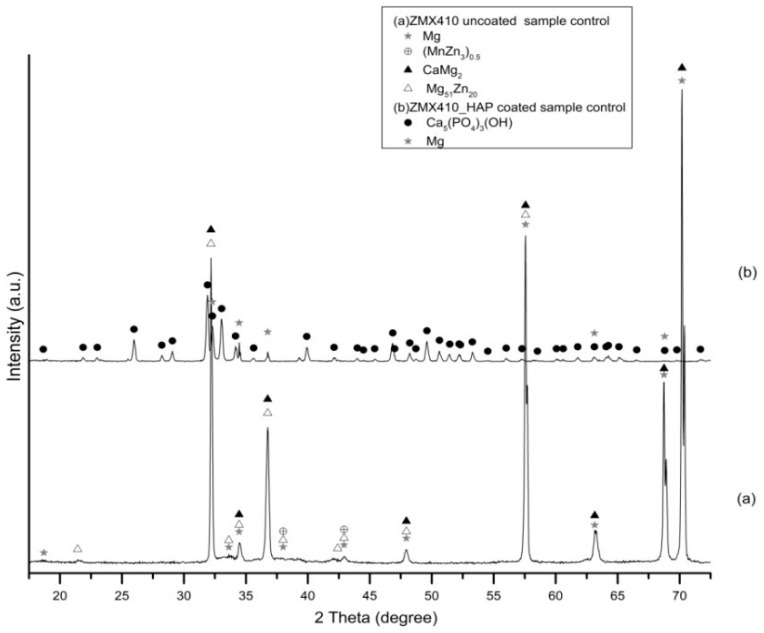
Patterns of uncoated (**a**) and hydroxyapatite (HAP)-coated (**b**) ZMX410 alloy (bullets indicate the peaks corresponding to diffraction angles of the main phases).

**Figure 7 materials-13-00263-f007:**
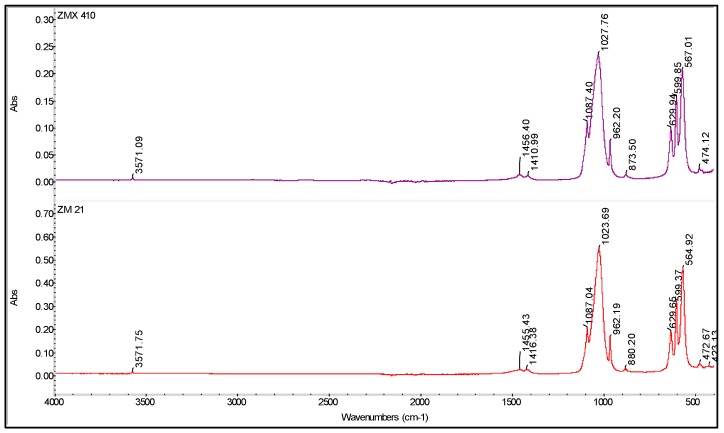
FTIR spectra of the coated ZM21 and ZMX410 samples.

**Figure 8 materials-13-00263-f008:**
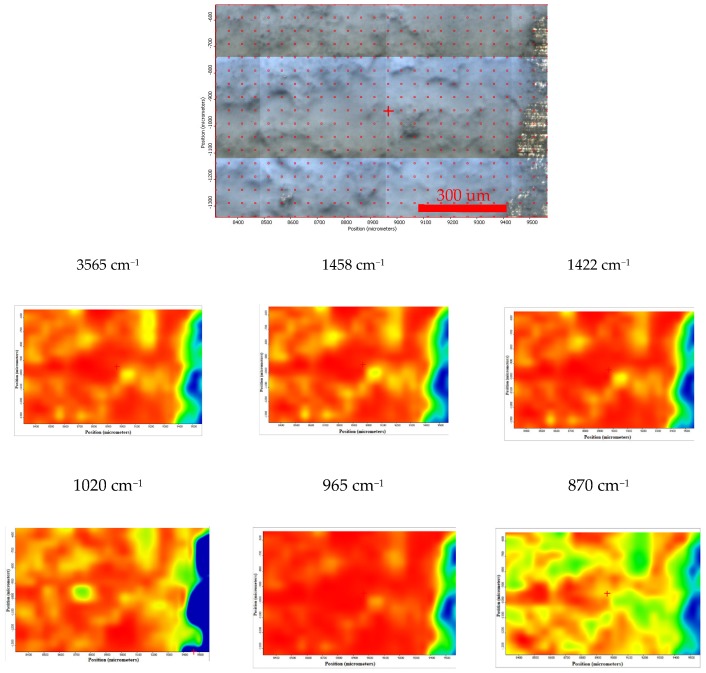
Chemigram maps of the coated ZM 21 alloy. In the upper part, a dark field optical image, corresponding to the investigated area, is presented. In the lower part, false colour maps with the indicated reference bands are shown. The chemigram scale bars are expressed in μm.

**Figure 9 materials-13-00263-f009:**
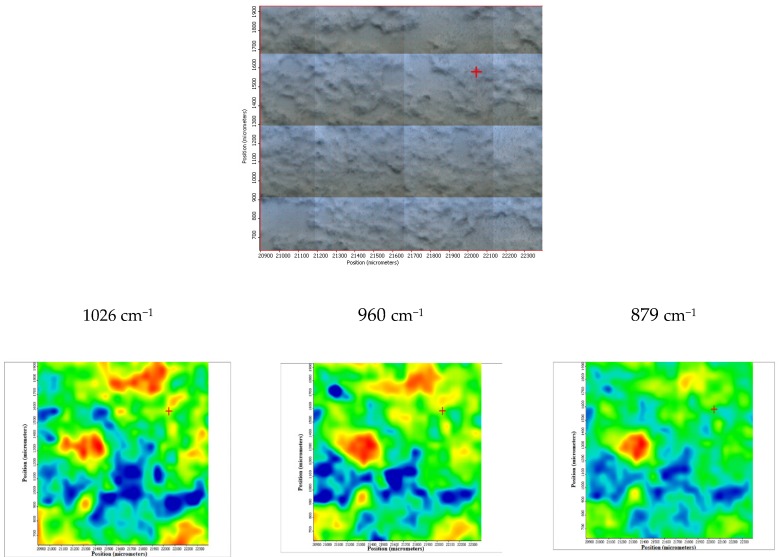
FTIR chemigram maps of the coated ZMX 410 alloy. In the upper part, a dark field optical image, corresponding to the investigated area, is presented. In the lower part, false colour maps with the indicated reference bands are shown. The chemigram scale bars are expressed in μm.

**Figure 10 materials-13-00263-f010:**
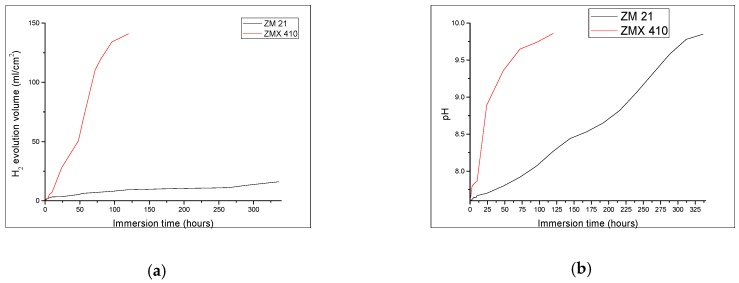
Behaviour of samples ZM21 and ZMX410 during immersion test: (**a**) The hydrogen release values registered for ZM21 and ZMX410 alloys; (**b**) pH values registered for ZM21 and ZMX410 alloys.

**Figure 11 materials-13-00263-f011:**
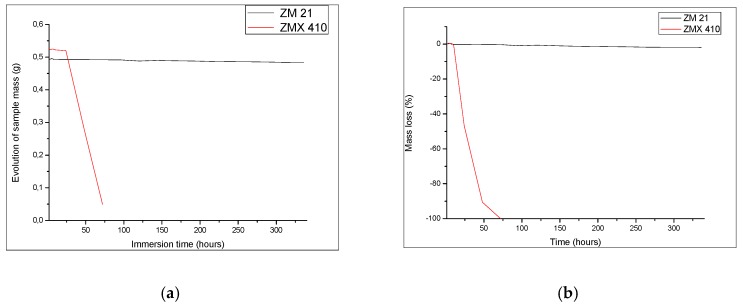
Quantitative behaviour of samples during immersion experiments: (**a**) variation of sample mass and (**b**) evolution of mass loss.

**Figure 12 materials-13-00263-f012:**
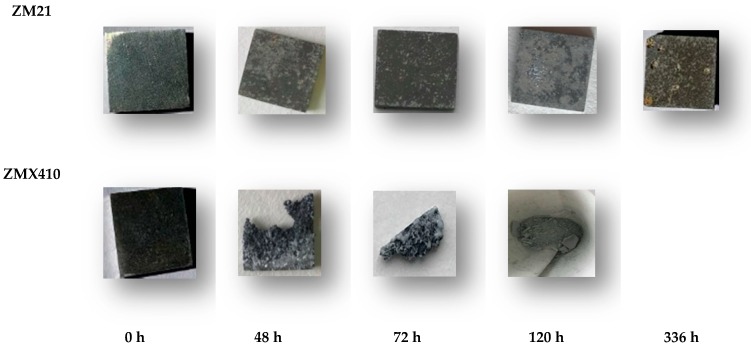
Aspects of the uncoated samples after immersion tests in SBF.

**Figure 13 materials-13-00263-f013:**
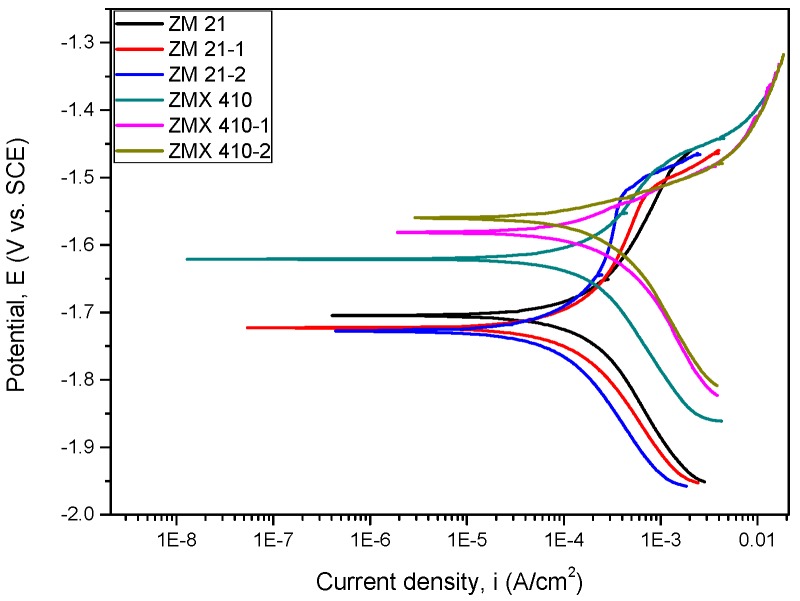
Open loop potential (E_oc_) for all samples tested in SBF.

**Figure 14 materials-13-00263-f014:**
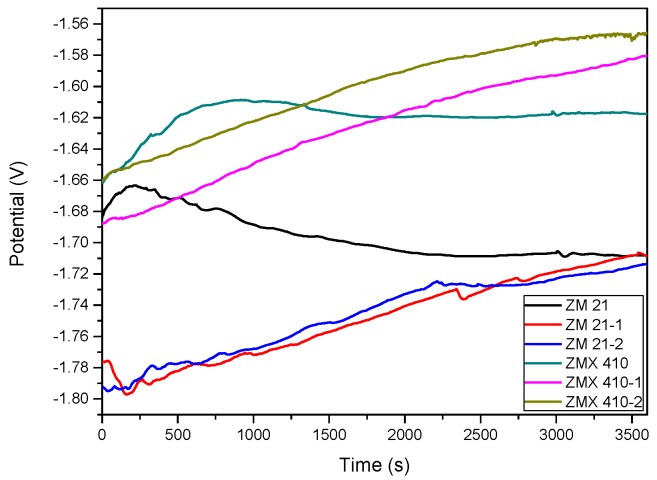
The Tafel curves of all samples tested in SBF.

**Figure 15 materials-13-00263-f015:**
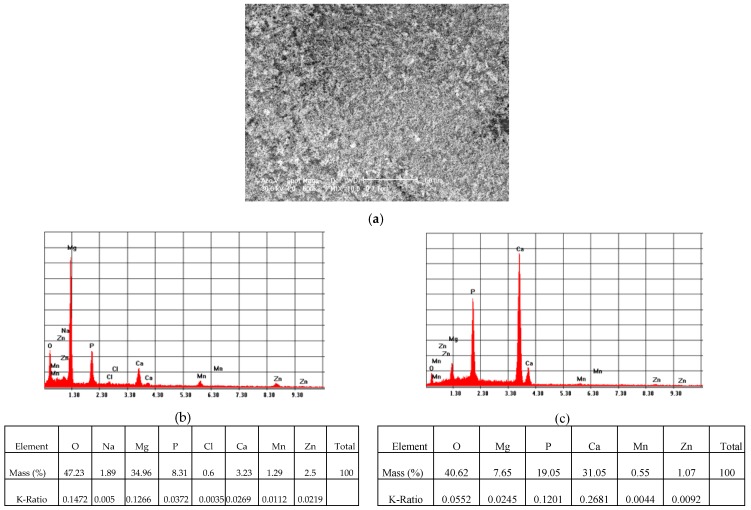
Representative SEM images (magnification 500×) of the HAP coated ZM21 sample (**a**), and corresponding EDS spectra (**b**) of corroded areas; (**c**) of uncorroded areas, after electrochemical corrosion in SBF.

**Figure 16 materials-13-00263-f016:**
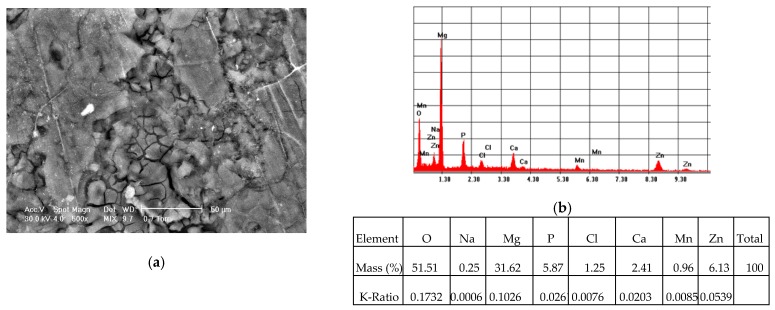
Representative SEM images (magnification 500×) of the HAP coated ZMX410 sample (**a**), and corresponding EDS spectrum (**b**) of corroded areas, after electrochemical corrosion in SBF.

**Figure 17 materials-13-00263-f017:**
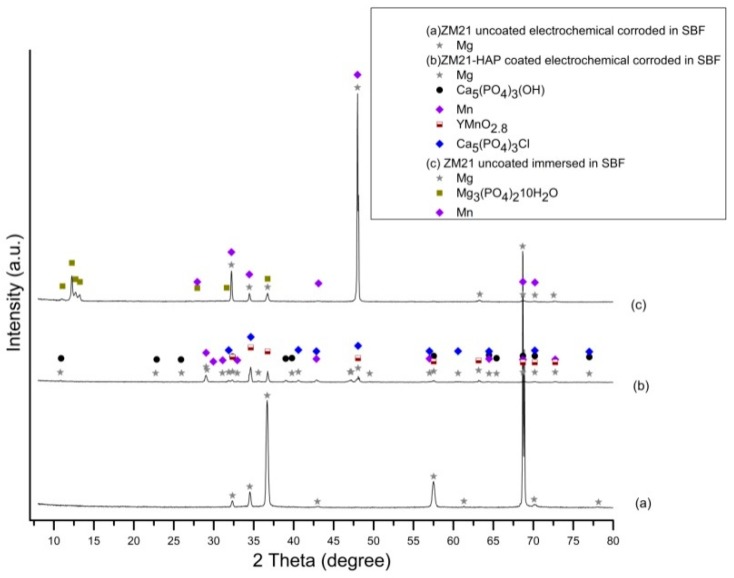
XRD patterns of uncoated ZM21 alloy after electrochemical corrosion (**a**), HAP coated ZM21 alloy after electrochemical corrosion (**b**) and uncoated ZM21 alloy after immersion corrosion in SBF (**c**) (bullets indicate the peaks corresponding to diffraction angles of the main phases).

**Figure 18 materials-13-00263-f018:**
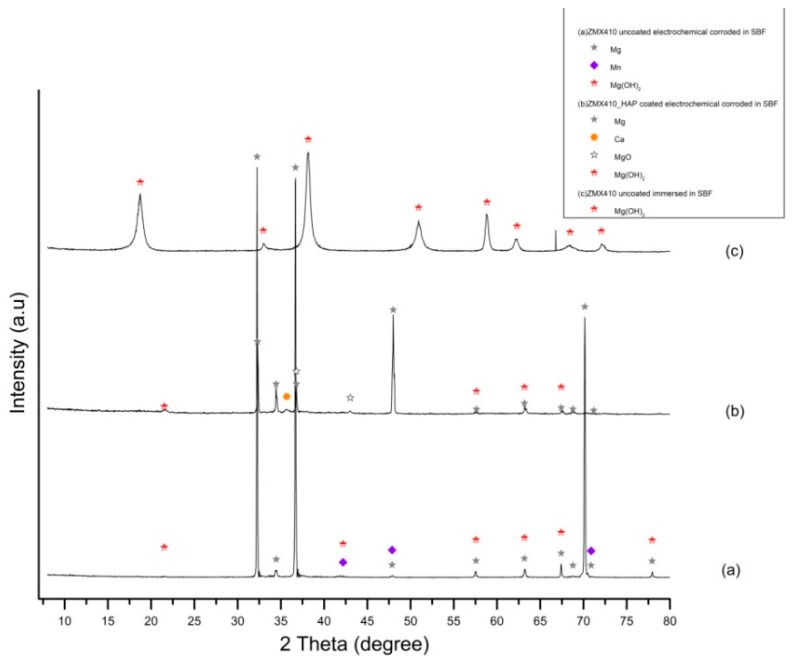
XRD patterns of uncoated ZM410 alloy after electrochemical corrosion (**a**), HAP coated ZM410 alloy after electrochemical corrosion (**b**) and uncoated ZM410 alloy after immersion corrosion in SBF (**c**) (bullets indicate the peaks corresponding to diffraction angles of the main phases).

**Figure 19 materials-13-00263-f019:**
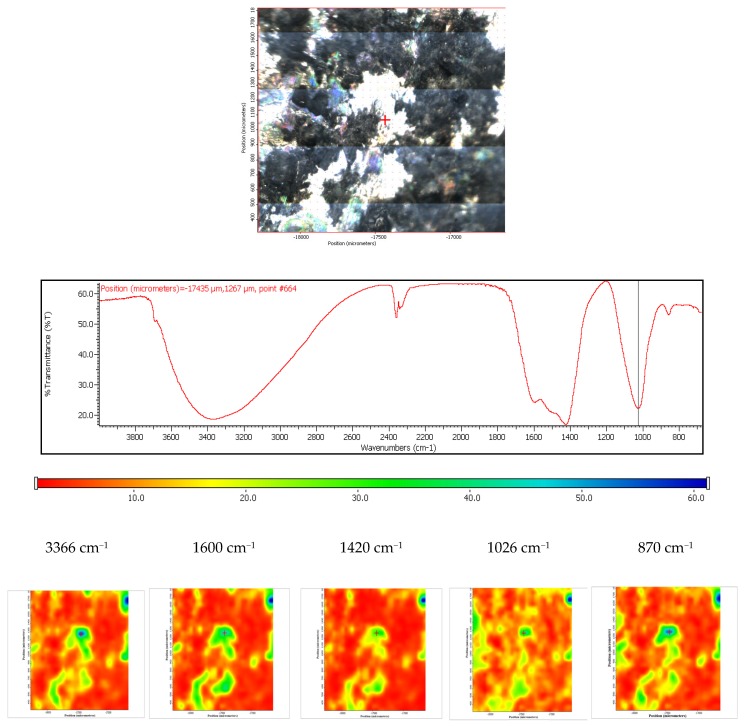
FTIR diagram and chemigram maps of the coated ZM21 alloy exposed to SBF. In the upper part, a dark field optical image, corresponding to the investigated area, is presented. In the lower part, false colour maps with the indicated reference bands are shown. The chemigram scale bars are expressed in μm.

**Figure 20 materials-13-00263-f020:**
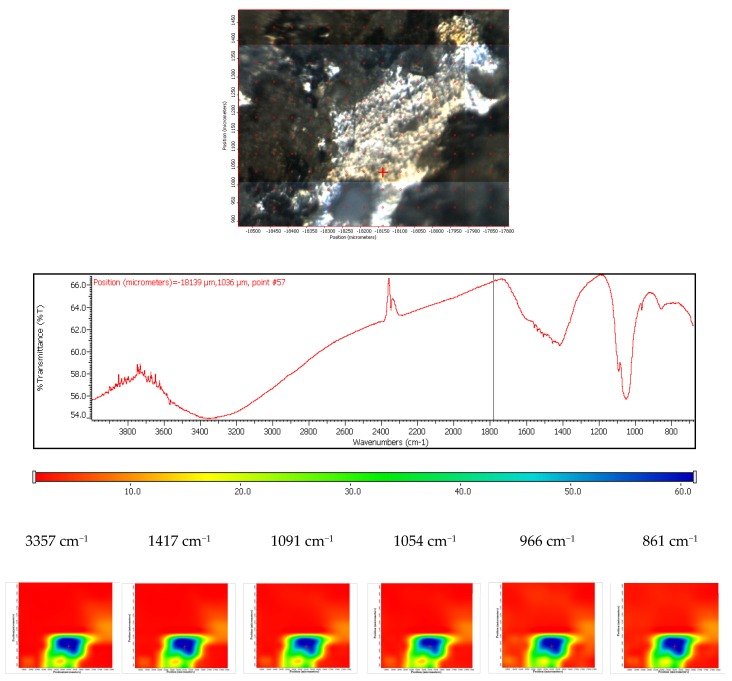
FTIR diagram and chemigram maps of the coated ZMX 410 alloy exposed to SBF. In the upper part, a dark field optical image, corresponding to the investigated area, is presented. In the lower part, false colour maps with the indicated reference bands are shown. The chemigram scale bars are expressed in μm.

**Figure 21 materials-13-00263-f021:**
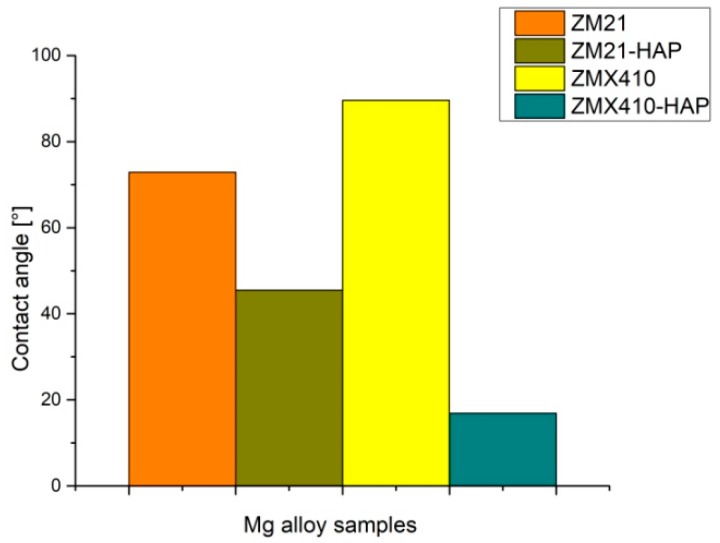
Water contact angles of uncoated and HAP coated ZM21 and ZMX410 alloys.

**Figure 22 materials-13-00263-f022:**
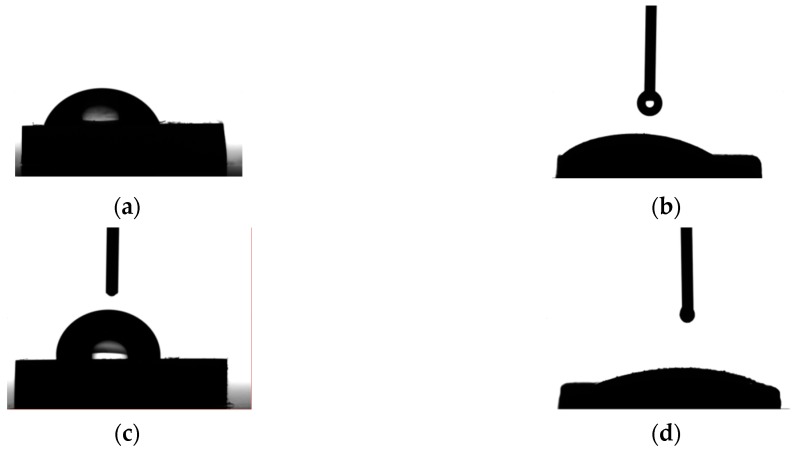
Photograph of water contact with Mg alloy surfaces: (**a**) ZM21; (**b**) ZM21-HAP; (**c**) ZMX410; (**d**) ZMX410-HAP.

**Table 1 materials-13-00263-t001:** Chemical composition of experimental Mg alloys.

Alloy	Chemical Element				
	Mg (%wt)	Zn (%wt)	Y (%wt)	Mn (%wt)	Ca (%wt)
ZM21	96.68	2.00	0.16	1.16	–
ZMX410	94.78	4.30	–	0.62	0.30

**Table 2 materials-13-00263-t002:** Corrosion rate of ZM21 and ZMX410 samples during the immersion test in SBF.

Time (h)	24	48	72	96	120	192	336
ZM21 CR (mm/year)	0.28	0.42	0.47	0.52	0.61	0.73	1.01
ZMX410 CR (mm/year)	37.81	38.58	39.77	–			

**Table 3 materials-13-00263-t003:** Main parameters determined during the corrosion process performed in SBF.

Sample	E_corr_ (V)	i_corr_ (µA/cm^2^)	β_c_ (mV)	β_a_ (mV)	CR (mm/year)	R_p_ (Ωxcm^2^)	P_e_ (%)
ZM21	−1.704	314.4	319.616	332.8	6.85	225.44	−
ZM21-1	−1.722	192.2	246.422	296.7	−	304.48	38.86
ZM21-2	−1.727	181.3	296.368	420.8	−	416.96	42.33
ZMX410	−1.620	295.7	290.676	236.2	6.34	191.56	−
ZMX410-1	−1.581	283.8	220.632	111.3	−	113.39	4.04
ZMX410-2	−1.560	282.6	233.624	90.8	−	100.62	4.42

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
