# Peer review of "Controlling the Degradation Rate of Biodegradable Mg–Zn-Mn Alloys for Orthopedic Applications by Electrophoretic Deposition of Hydroxyapatite Coating"

_materials, 2020, doi:10.3390/ma13020263_

Round 1

Reviewer 1 Report

Thank you for the manuscript. The study describes the effects of coating the Mg-Zn-Mn alloy surface with hydroxyapatite layer in order to evaluate the corrosion resistance and biodegradation rate of tested alloys. The paper is interesting and quite well written. In some points of Introduction section, too long sentences lose their sense. Study seems well designed and performed. However, the title suggests that different thicknesses of the HAp coatings would be tested to control the degradation rate of HAp coated Mg-Zn-Mn alloys. The experimental part impresses with variety of methods used to describe the HAp coating. The discussion part is non-existent. Conclusions section should be shortened and pointing the most important issues of the study. Purely ornamental sentences such as “SEM analysis offered microscopical aspects of the coated and uncoated ZM21 and ZMX410 alloys prior and after the corrosion experiments.” should be removed.

Here are some issues the authors should address:

Abstract: line 24: remove “magnesium” as it is repetition of Mg-Zn-Mn

What mechanical properties were tested in the study?

Keywords: please, split the phrase “biodegradable magnesium alloys Mg-Zn-Mn” into few keywords

line 40: what does “ZM” mean?

line 48: should be “apart from”.  instead of “apart of”

Please rewrite the sentences in lines 48-51 and lines 51-54 as they are too long and hard to follow.

line 63: should be “the” instead of “those”

line 71-71: “They are harmless for the human body, this is why, a few years after commercial production, Mg implants, such as screws, sheets and yarns started to be used in surgical procedures” What do you mean? Did they need few years of commercial production to pass before they could be used in surgical procedure? Please, rewrite the sentence.

line 76: put “in vivo” in italics

The aim of the study should be clearly stated at the end of introduction section.

Table 1: correct the values so they have the same number of decimal places

lines 150-151: rewrite the sentence, as it is misleading: “for selected periods of time 30 minute”, “ultrasonic treatment at ultrasonic intensity”?

line 224 vs. line 260 vs. Table 2: corrosion rate - RC or CR?

line 238: what do you mean by “SBF tests”? Please, rephrase.

lines 284-287: please add references (ref# 60 is not suitable here)

line 290:  correct the spelling of the names, do not use uppercase letters

Fig. 1 and Fig 2: please, present figures taken at the same magnification otherwise it gives confusing information. There is a significant difference in mag. 20000x, 2000x and 500x

Also, please explain how the samples were produced for cross section observation, since there was no mention of it in the Method section (paragraph 2.2.1.)

line 317: “HAP layer of approximately 15-16 µm thickness was observed” How was it measured? no description in methods is available.

Fig. 4: please put in the description, which EDS belongs to which point indicated by arrow, e.g. (b) globular particles (blue arrow)

lines 388-391: to long sentence, please, split it in two

Fig. 10: Please add description of (a) and (b)

line 403: should be “was investigated”

Figure 12 is of a poor quality

Table 3: use one type of decimal separator

line 436: “relative to”? perhaps “comparing to/in comparison to”

lines 435-439: “Electrochemical measurements showed that the values obtained for the HAP coated ZM21 alloy are approximately equal but less electropositive relative to the Mg alloy substrate. The difference between these values is, however, quite small, only 18 mV for the ZM 21-1 and 23mV for the ZM 21-2. The HAP coatings on the ZM 21 alloy have resulted in corrosion potentials more electropositive than the substrate, and, thus, have a better corrosion resistance behavior.” So which is it?

“The difference between these values.” which values?

line 552: “to protect the metal substrate” in what way?

Discussion: Corrosion resistance of tested Mg alloys was calculated. What about the CR results of HAp coated Mg alloys?

lines 571-572: ”Their corrosion rate is within acceptable limits, which allows us to recommend them for medical devices with temporary placement.” what are the acceptable limits? any references?

The abovementioned issues do not substantially decrease the value of the paper.

Author Response

Here are the answers to the comments made by the reviewer 1:

Many thanks for your valuable comments. The authors answer to all comments, as is mentioned below.

Abstract: line 24: remove “magnesium” as it is repetition of Mg-Zn-Mn. What mechanical properties were tested in the study?

We modify the abstract, because the mechanical properties were not tested in this paper.

Keywords: please, split the phrase “biodegradable magnesium alloys Mg-Zn-Mn” into few keywords

We made the suggested modification.

line 40: what does “ZM” mean?

ZM mean that the main alloying elements of magnesium based alloys are Z = zinc and M = manganese. Anyway, we appreciate this comment and we use the term Mg-Zn-Mn alloys instead of Mg-Zn alloys in the entire manuscript.

line 48: should be “apart from”.  instead of “apart of”. / Please rewrite the sentences in lines 48-51 and lines 51-54 as they are too long and hard to follow.

We rewrite the sentences.

line 63: should be “the” instead of “those”

Done.

line 71-71: “They are harmless for the human body, this is why, a few years after commercial production, Mg implants, such as screws, sheets and yarns started to be used in surgical procedures” What do you mean? Did they need few years of commercial production to pass before they could be used in surgical procedure? Please, rewrite the sentence.

We delete the sentence.

line 76: put “in vivo” in italics

Done.

The aim of the study should be clearly stated at the end of introduction section.

We add some sentences about the aim of the study in the introduction section.

(line 87-94 in revised manuscript)

Table 1: correct the values so they have the same number of decimal places

We correct the values.

lines 150-151: rewrite the sentence, as it is misleading: “for selected periods of time 30 minute”, “ultrasonic treatment at ultrasonic intensity”?

The sentence was rewritten.

line 224 vs. line 260 vs. Table 2: corrosion rate - RC or CR?

The correct term is corrosion rate. So, the CR is appropriate abbreviation. In the manuscript were made the proper modifications.

line 238: what do you mean by “SBF tests”? Please, rephrase.

We made the suggested modifications.

lines 284-287: please add references (ref# 60 is not suitable here)

Done.

line 290:  correct the spelling of the names, do not use uppercase letters

Done.

Fig. 1 and Fig 2: please, present figures taken at the same magnification otherwise it gives confusing information. There is a significant difference in mag. 20000x, 2000x and 500x

We use other images in figure 1, in order to have a similar magnification. In the figure 2, the images were at the same magnification. Anyway, we consider that is better to select relevant SEM images.

Also, please explain how the samples were produced for cross section observation, since there was no mention of it in the Method section (paragraph 2.2.1.)

Done.

line 317: “HAP layer of approximately 15-16 µm thickness was observed” How was it measured? no description in methods is available.

It was measured using the SEM software.

Fig. 4: please put in the description, which EDS belongs to which point indicated by arrow, e.g. (b) globular particles (blue arrow)

We perform the suggested modifications (e.g. blue, white & red arrow).

lines 388-391: to long sentence, please, split it in two

We split the sentence.

Fig. 10: Please add description of (a) and (b)

The description was added.

line 403: should be “was investigated”

Done.

Figure 12 is of a poor quality

We use the plain images for new figure 12.

Table 3: use one type of decimal separator

Done.

line 436: “relative to”? perhaps “comparing to/in comparison to”

Thank you for your suggestion. The correction was made in the revised manuscript.

lines 435-439: “Electrochemical measurements showed that the values obtained for the HAP coated ZM21 alloy are approximately equal but less electropositive relative to the Mg alloy substrate. The difference between these values is, however, quite small, only 18 mV for the ZM 21-1 and 23mV for the ZM 21-2. The HAP coatings on the ZM 21 alloy have resulted in corrosion potentials more electropositive than the substrate, and, thus, have a better corrosion resistance behavior.” So which is it?

Thank you. The paragraph was modified as follow: “Electrochemical measurements showed that the corrosion potential values obtained for the ZM21 alloy uncoated and HAP coated are approximately equal,with quite smalldifference: only 18 mV for the ZM 21-1 and 23mV for the ZM 21-2. Even that the corrosion potential of HAP coated ZM21 alloy are less electropositive (with just a few mV) comparing to the Mg alloy substrate, an overview of all electrochemical parameters must be made prior to a conclusion.”

“The difference between these values.” which values?

Done.

line 552: “to protect the metal substrate” in what way?

We replace these sentence.

Discussion: Corrosion resistance of tested Mg alloys was calculated. What about the CR results of HAp coated Mg alloys?

To calculate the corrosion rate of a specimen with the help of electrochemical techniques, important parameters of material should be considered like density and equivalent weight. In case of coated specimens two different materials are immersed and the CR cannot be calculated.

lines 571-572: ”Their corrosion rate is within acceptable limits, which allows us to recommend them for medical devices with temporary placement.” what are the acceptable limits? any references?

We modify this sentence from conclusion. The new sentence is: Based on the experimental results obtained we consider that the degradation rate of biodegradable Mg–Zn-Mn alloys for temporary orthopedic applications could be controlled by electrophoretic deposition of hydroxyapatite coating [11,18,68,69].

Reviewer 2 Report

The manuscript is already in good shape, but still needs to be improved. The changes suggested, however, are mainly of technical character.

1) To facilitate reading, on page 3, last paragraph of Introduction section, it is recommended to add 1-2 sentences that clearly mention what new and original the present work presents in comparison to others.

2) The use of units should be checked and unified. For example, on page 4, one can see both "minutes" (at least 4 times) and "min" (at least 1 time). for consistency, it is better to use same style.

3) Through the text, spelling of some techniques used should be slightly corrected. For example, line 304:    Scanning electron microscopy and energy -dispersive spectroscopy have some words spelled with capital letters (while the others spelled with low-case letters). Similar inconsistency can be found all through the text. Same concerns line 331 (X-ray Diffraction results)

4) Fig.2, especially panels (b) and (d): scale bars are not well seen and probably should be improved (added?).

5) This reviewer feels that the manuscript is somewhat over-loaded with figures (some of which are not discussed extensively). Such figures, optionally, could be moved to Supplementary Information. In the text, they can be mentioned in words, while referring the reader to the Supplementary Information file. This is believed to make the article slightly shorter and easier to read (when less important figures disappear and let the reader focus on the text and on the key figures). There are ,for example, quite many XRD patterns that are not well seen (having many details) but not discussed extensively. Maybe some of them can be removed (?). This comment is quite optional, and I would let the authors decide.

6) The reference list appears to need changes. Author names in all the references are presented improperly: their initials should follow their surnames. The authors should check the standard/requirements used by Materials for reference list.

Author Response

Here are the answers to the comments made by the reviewer 2:

Many thanks for your valuable comments. The authors answer to all comments, as is mentioned below.

1) To facilitate reading, on page 3, last paragraph of Introduction section, it is recommended to add 1-2 sentences that clearly mention what new and original the present work presents in comparison to others.

We introduce new sentences in revised manuscript.

2) The use of units should be checked and unified. For example, on page 4, one can see both "minutes" (at least 4 times) and "min" (at least 1 time). for consistency, it is better to use same style.

We made the modifications in order to unify the use of units.

3) Through the text, spelling of some techniques used should be slightly corrected. For example, line 304:    Scanning electron microscopy and energy -dispersive spectroscopy have some words spelled with capital letters (while the others spelled with low-case letters). Similar inconsistency can be found all through the text. Same concerns line 331 (X-ray Diffraction results)

We made the suggested modifications in revised manuscript.

4) Fig.2, especially panels (b) and (d): scale bars are not well seen and probably should be improved (added?).

We made the suggested modifications in revised manuscript, using proper images.

5) This reviewer feels that the manuscript is somewhat over-loaded with figures (some of which are not discussed extensively). Such figures, optionally, could be moved to Supplementary Information. In the text, they can be mentioned in words, while referring the reader to the Supplementary Information file. This is believed to make the article slightly shorter and easier to read (when less important figures disappear and let the reader focus on the text and on the key figures). There are ,for example, quite many XRD patterns that are not well seen (having many details) but not discussed extensively. Maybe some of them can be removed (?). This comment is quite optional, and I would let the authors decide.

Many thanks. We will try to follow this suggestion for our future articles.

6) The reference list appears to need changes. Author names in all the references are presented improperly: their initials should follow their surnames. The authors should check the standard/requirements used by Materials for reference list.

We made the suggested modifications in revised manuscript.

Round 2

Reviewer 1 Report

Thank you for introducing all the suggested changes into the manuscript.

One minor issue:

Lines 156-158: “In order to evaluate coating uniformity over the magnesium alloys substrates, SEM was used to measured, at various locations, coating thickness. The experimental samples were prepared using argon ion beam in order to see the cross-section of the coating layer”

Please,  specify “various locations”; was it a mean value of the measurements taken at few  different spots?

Author Response

Thank you for the comment.

Just in order to clarify this minor issue, I would like to mention that we select an area that correspond to the mean value obtained after 5 measurement in different spots. We shown just the relevant SEM image in the manuscript.

In general term of debates, our research group perform HAP coatings on Mg based alloys using various methods. We found that electrophoretic deposition assure the best homogeneous and uniform HAP layer on the Mg alloys substrates, comparing to other coating methods, but this is a topic for a future paper.

I hope that the reviewer will agree our comments and the manuscript will be published as was submitted after first revision.

Reviewer 2 Report

The manuscript was improved and now can be accepted for publication

Author Response

Many thanks for your appreciation.